# RF Interconnection Design of Bump Bonding with a Dislocation Package Structure towards Electro-Optic Modulation Applications

**Jiahao Peng** [1,2]**, Xiaofeng Wang** [1,]***, Libo Wang** [2]**, Yang Li** [2]**, Runhao Liu** [2]**, Shiyao Deng** [2]**, Heyuan Guan** [2]
**and Huihui Lu** [1,2,]***

1    SJTU-Pinghu Institute of Intelligent Optoelectronics, Pinghu 314200, China
2    Guangdong Provincial Key Laboratory of Optical Fiber Sensing and Communications, Jinan University, Guangzhou 510632, China
*    Correspondence: xiaofeng.wang@spioe.cn (X.W.); thuihuilu@jnu.edu.cn (H.L.)

**Abstract:** Bonding technology can be an important component of packaging for photonic chips, such as electro-optic (EO) modulators and other active function devices. In general, an EO modulator chip can achieve a broader 3 dB EO bandwidth than its packaging device, as the packaging design and structure can technically limit modulation performance. Recently, bump bonding has been shown to be a good candidate for the EO interconnection technique, which has a higher transmission bandwidth than wire bonding. In this article, we propose a design for radio frequency (RF) interconnection of bump bonding with a dislocation packaging (BBDP) structure. Through simulation calculations and analysis, the proposed BBDP structure shows a 3 dB transmission bandwidth of approximately 145 GHz, which is 52.6% better than one using optimized wire-bonding structures (95 GHz). The proposed packaging structure presents an important alternative method for ultrahigh speed optical modulation applications.

**Keywords:** RF interconnection; modulator; bump bonding; dislocation packaging

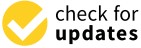



## 1. Introduction

An electro-optical modulator is a vital device in an optoelectronic module. It can convert electrical signals into optical signals and perform low-loss transmission through optical fibers. It is widely utilized in optical fiber communications, optical fiber sensing, nanophotonics, microwave photonics, and other technical fields [1–6]. An RF transmission line such as microstrip can form the travelling-wave electrodes of an electro-optic (EO) modulator and is the key platform for wire bonding, bump bonding, and other packaging techniques [6,7]. Its function is to convey the RF signal from the RF connector to the electrodes of the EO modulator or other EO interconnection devices to achieve the conversion from electric signal to optical signal [8–10]. Common methods for interconnecting these two devices are wire bonding or bump bonding [11]. Generally, bump bonding has better packaging performance than wire bonding [12], which is useful and available in the current packaging structure [13–15].

During the packaging process of a modulator, it is usually necessary to fabricate a 90° bent transition for the electrode structure at the input end and output end of the modulator electrode [16,17]. The purpose of this is to extend the electrode structure on the sides of the modulator chip and to interconnect the EO modulator and the RF transition chip through wire bonding. The 90° curved electrode structure brings considerable bandwidth loss to the device.

In order to improve the transmission performance of an RF modulator interconnection, we mainly optimize two aspects of the design. On the one hand, we use bump bonding to

achieve RF interconnections instead of wire bonding, which induces impedance mismatching. On the other hand, we should avoid a design with curved electrodes to further reduce loss during RF transmissions.

Based on these two directions, we proposed and designed a dislocation packaging solution for RF interconnections in an EO modulator. In our design, the interconnected devices avoid the bending structure of the electrodes through misalignment design. RF transition chips are located above the EO modulator and are perpendicular to each other. At the same time, the two misaligned devices are interconnected through bump bonding. Therefore, the electrode structure of the modulator does not need to be bent at 90°, and the performance advantages of bump bonding can also be used, which can effectively reduce microwave losses and increase the transmission bandwidth.

In this article, we first introduce the structure we designed in detail and use the finite element method (FEM) to simulate the model using ANSYS HFSS software to obtain its S parameter data. Subsequently, attempts were made to expand the designed structure to be multi-channel, and the feasibility of this interconnection scheme in multi-channel modulators was explored. Finally, we simulated and compared the EO modulator RF interconnection and wire-bonding models with a 90° bent electrode structure. By comparing the wire-bonding structure, our proposed BBDP structure shows a 3 dB transmission bandwidth of approximately 145 GHz, which is 52.6% better than the wire-bonding structure (95 GHz). This presents an important alternative path to realizing a high-speed optoelectronic chip, such as an EO modulator, photodetector, etc.

## 2. RF Interconnection Design of Bump Bonding with a Dislocation Package Structure

Compared with wire bonding, bump bonding can effectively reduce the parasitic capacitance brought by the wire: its interconnection length is greatly shortened, the delay of RC resistance and capacitance is reduced, and its electrical performance is effectively improved. Based on bump bonding, we designed a dislocation packaging structure for a modulator RF interconnection. The overall structure is shown in Figure 1.

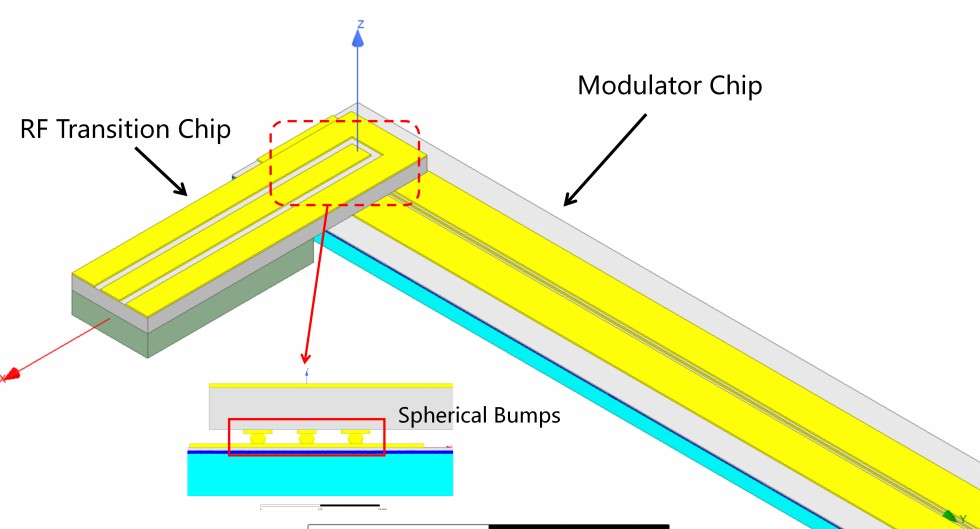

**Figure 1.** Overall structure of the BBDP structure.

The structure consists of a modulator chip, an RF transition chip, and spherical bumps for bonding between them. As shown in Figure 1, the RF transition chip designed in this paper is misaligned with the modulator in the z-axis direction (non-coplanar structure), and the electrode structure of the modulator does not need to be provided with a 90° curved electrode structure. The radio frequency transition chip is perpendicular to the modulator on the x–y plane. The RF transition chips are located on top of the modulator, and they are perpendicular to each other in the x–y plane. Figure 2 and Table 1 show detailed parameters of the BBDP structural design.

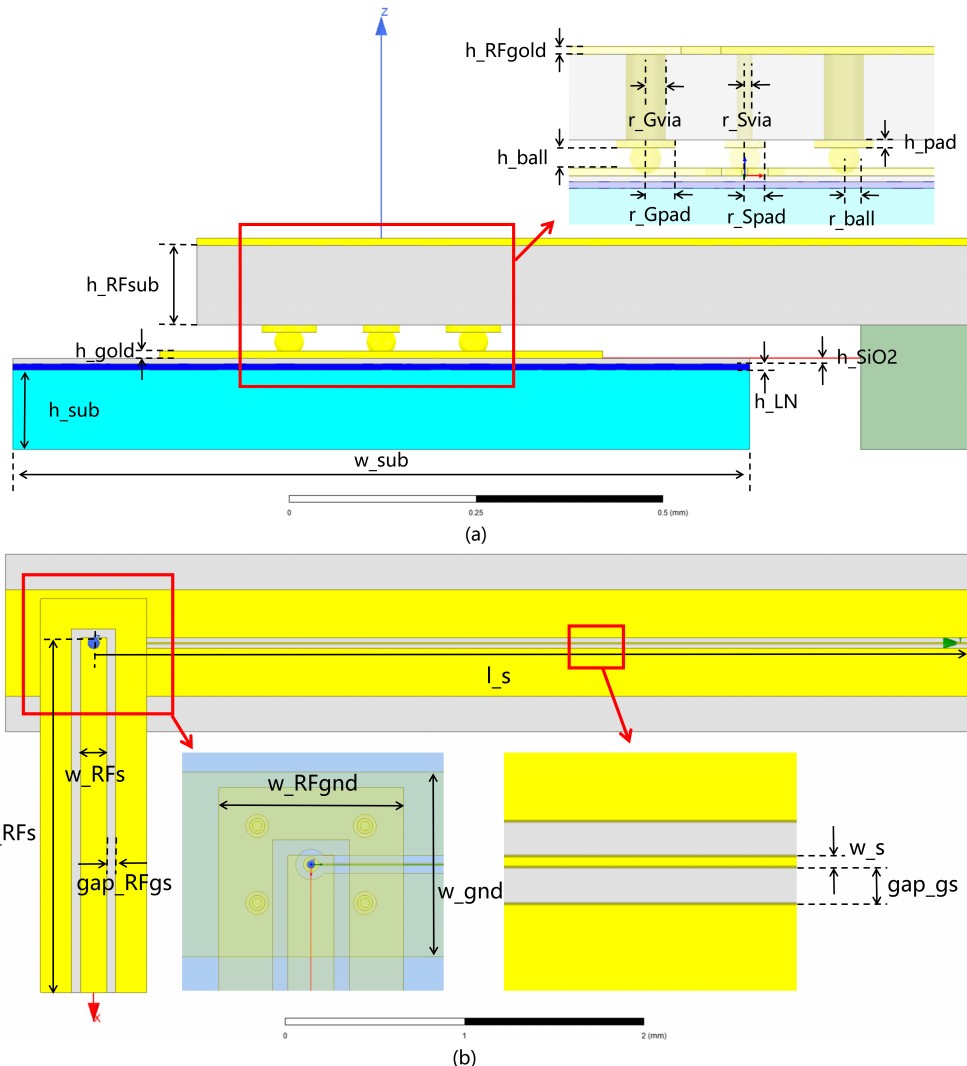

**Figure 2.** (**a**) x–z section view of BBDP structure; (**b**) x–y top view of BBDP structure.

In the structure in Figure 2, we used a lithium niobate electro-optical modulator as a demonstration. The material of the electrode structure was gold with a thickness of 10 μm. The signal electrode consisted of a circular pad with a radius of 25 μm and a rectangular electrode with a width of 8 μm. The ground electrodes on both sides of the signal electrode had a width of 371 μm, the same thickness as a signal electrode. The distance between the ground electrode and the signal electrode (gap_gs) was 25 μm.

The structure of an RF transition chip from top to bottom is the coplanar waveguide GSG (ground–signal–ground) electrode structure with alumina substrate, metal vias, and pads, as illustrated in Figure 2. The material of the electrode structure, the vias and pads, was gold. Specifically, the signal electrode was 150 μm wide and 10 μm thick, and the ground electrodes on both sides of the signal electrode had a width of 175 μm, the same thickness as the signal electrode. The distance between the ground electrode and the signal electrode was 50 μm, and the thickness of the alumina substrate was 108 μm. There were several metal pads distributed on the bottom of the alumina substrate for connecting the spherical bumps: the radius of the signal pad was 30 μm, and the radius of the ground pad was 40 μm. The electrode structure above the substrate was connected to the pads below the substrate through metal vias. The radius of the signal via was 20 μm, and the radius of the ground via was 30 μm. Under the RF transition chip and outside of the LN electro-optic modulator chip, there was a quartz block to provide mechanical support for the radio frequency transition chip, as indicated in Figure 2.

**Table 1.** Specific values of the structural parameters shown in Figure 2.

| Parameter Symbols | Value (μm) | Parameter Symbols | Value (μm) |
|---|---|---|---|
| h_RFgold | 10 | r_Gvia | 30 |
| r_Svia | 10 | h_pad | 10 |
| r_Gpad | 40 | r_Spad | 25 |
| h_ball | 30 | r_ball | 25 |
| h_RFsub | 108 | h_gold | 10 |
| h_SiO$_2$ | 10 | h_LN | 10 |
| h_sub | 108 | w_sub | 1200 |
| w_RFgnd | 600 | l_RFs | 2000 |
| w_RFs | 150 | gap_RFgs | 50 |
| w_gnd | 800 | l_s | 3000 |
| w_s | 8 | gap_gs | 25 |

The modulator and the RF transition chip were bump-bonded through golden balls. The radius of the golden ball was 25 μm and the height was 30 μm. For configurating the bump bonding of ground electrodes, multiple golden balls could be used to improve the stability of the device interconnection and provide certain auxiliary mechanical support for the transition chip. Here we simply used four gold balls for the bump bonding of the ground electrode.

In high-speed CPW traveling wave modulators, RF transmission line characteristics, such as impedance matching, microwave losses, and group velocity matching of microwaves and light waves, must be considered. In this article, we mainly focussed on the scattering parameters (S parameters) of the device model, which are closely related to the characteristic impedance ($Z_0$). The characteristic impedance is related to the resistance (R), inductance (L), and capacitance (C) of the CPW electrode per unit length. The relationship between R, L, G, C and $Z_0$ is as shown in the following formula [18]:

$$Z_0 = \sqrt{\frac{R + j\omega_{RF}L}{G + j\omega_{RF}C}} \tag{1}$$

The obtained characteristic impedance can be used to further calculate the S parameter. When the input characteristic impedance $Z_0$ is 50 ohms, we can calculate the reflection coefficient R and transmission coefficient T:

$$R = \frac{Z_c - Z_0}{Z_c + Z_0} \tag{2}$$

$$T = e^{-\gamma L} \tag{3}$$

$$\gamma = \sqrt{(R + j\omega_{RF}L)(G + j\omega_{RF}C)} \tag{4}$$

where $\gamma$ is the propagation constant. S parameters can be obtained:

$$S_{12} = S_{21} = \frac{T(1 - R^2)}{1 - R^2T^2} \tag{5}$$

$$S_{11} = S_{22} = \frac{R(1 - T^2)}{1 - R^2T^2} \tag{6}$$

In this work, we simulated the CPW electrode and analyzed the transmission characteristics of the electrode. We used FEM to simulate the structure and calculate its S parameters to evaluate the RF performance.

FEM is a numerical algorithm that needs to discretize the propagation space. The absorption boundary condition is a necessary boundary condition that is used to simulate the natural propagation state of electromagnetic waves, which is similar to the principle of microwave anechoic chamber simulating electromagnetic wave propagation in free space. We placed our model in an air box with radiative boundary conditions. The radiation boundary also refers to the absorption boundary. The system absorbed incoming waves at the radiation boundary, essentially extending the boundary to infinity to achieve the simulation of free space. If the free space is equivalent to a waveguide, the radiation boundary is equivalent to the matching load at the end of the waveguide.

To initially evaluate the advantages of our design, we conducted supplementally calculations. Figure 3 shows the microstrip interconnection model using wire bonding and dislocation bump bonding, respectively. In Figure 3a, two microstrip lines are bonded through gold wires. In Figure 3b, two microstrip lines are bonded through via and bump. The microstrip lines, substrate, and simulation settings were identical in both models. Figure 4 illustrates the S parameter results for comparison, and Figure 5 shows the port impedance curve. Figure 5a is the impedance curve at port 1, and Figure 5b is the impedance curve at port 2. Port 1 and port 2 are the input and output ports at both ends of the model respectively. It can be seen that the use of dislocated bump bonding effectively improved the transmission bandwidth of the microstrip line. In the simulation, we used a 50-ohm wave port as the excitation source. Therefore, as mentioned above, the model's port characteristic impedance was approximately close to 50 ohm, and a better transmission performance was obtained. As shown in Figure 5, compared to wire bonding, dislocation bump bonding could achieve better impedance matching, which could more effectively reduce transmission loss and increase transmission bandwidth.

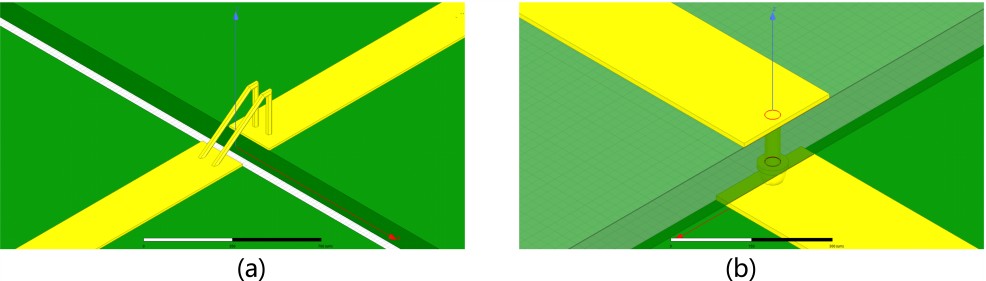

(a) (b)

**Figure 3.** Interconnected simulation model of two bonding methods.

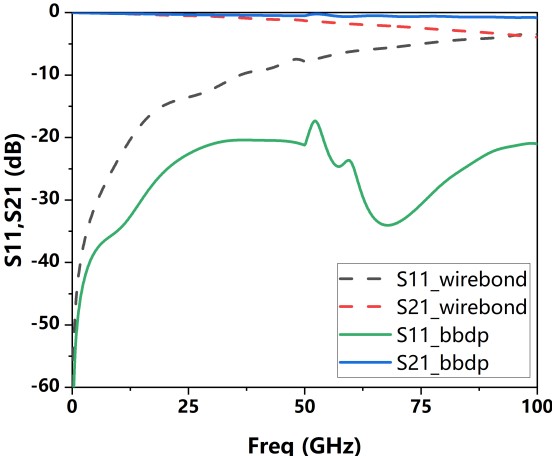

**Figure 4.** S parameter curves of the two methods.

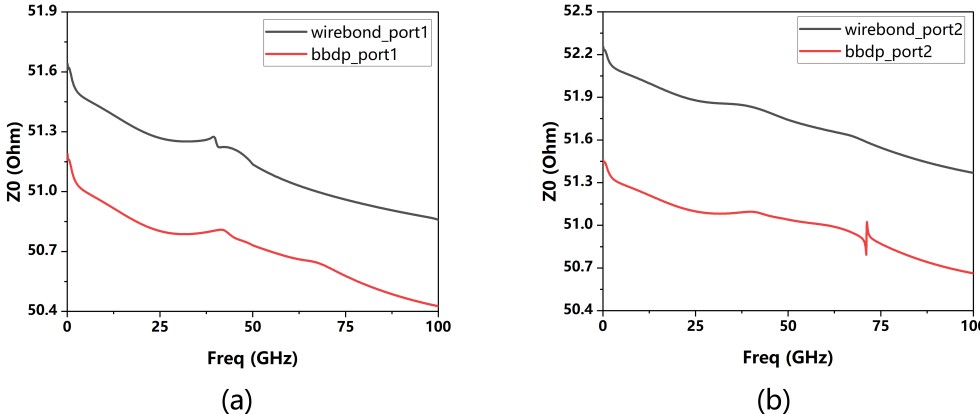

**Figure 5.** Port characteristic impedance of the two models.

Through the above simulation comparison, we believe that it is reasonable to use dislocated bump bonding for packaging to improve transmission performance. The following is a description of the simulation setup for the model shown in Figure 2. The solution type was assigned to the terminal type and the overall model was placed in an air box, where it was assigned to the radiation boundary condition. The ground electrodes were assigned to the perfect E boundary condition, whereas the excitation sources of the signal electrode of the RF transition chip and the signal electrode of the modulator were both assigned to lump port excitation. The sweep frequency range was set up to an interpolated frequency range from 0 to 150 GHz. The simulation results of the S parameters (S11, S21) are shown in Figure 6. The 3 dB transmission bandwidth (S21) of this structure was about 145 GHz.

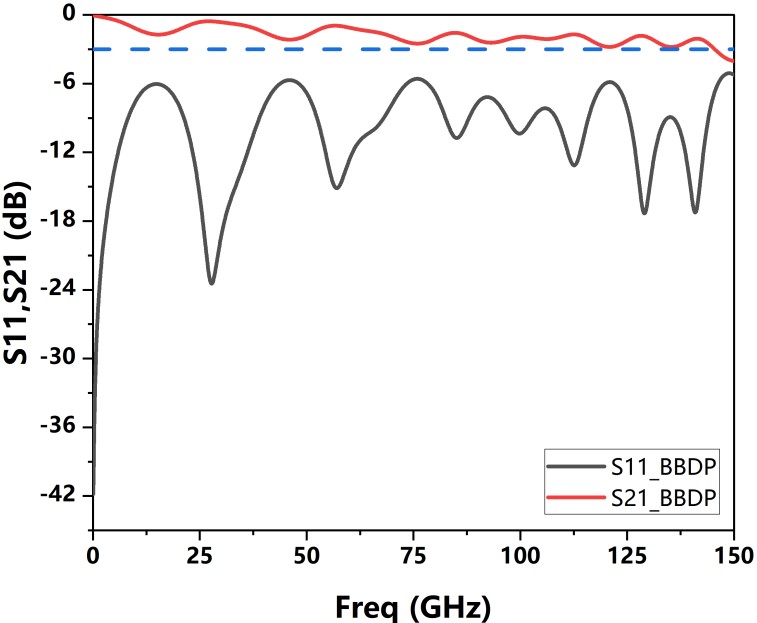

**Figure 6.** S parameter simulation results of proposed BBDP structure.

Furthermore, the above-proposed structure could be further extended to the design of the dual-channel RF interconnection of bump bonding with a dislocation packaging structure, as shown in Figure 7. In this case, there were two modulation arms, where the two signal electrodes of the modulator in this structure shared a common ground electrode, forming two pairs of GSG coplanar waveguide electrode structures. The parameters of this structure were consistent with the structure shown in Figure 2. It should be added that the spacing between the signal electrodes in the RF transition chip was 100 μm, the width of the modulator ground electrode was 750 μm, and the spacing between the signal electrodes

was 300 μm. Similarly, the two signal electrodes of the RF transition chip were distributed on top of both sides of the modulator and shared the ground electrode, forming two sets of GSG coplanar waveguide structures. The simulated S parameters of this structure are shown in Figure 8.

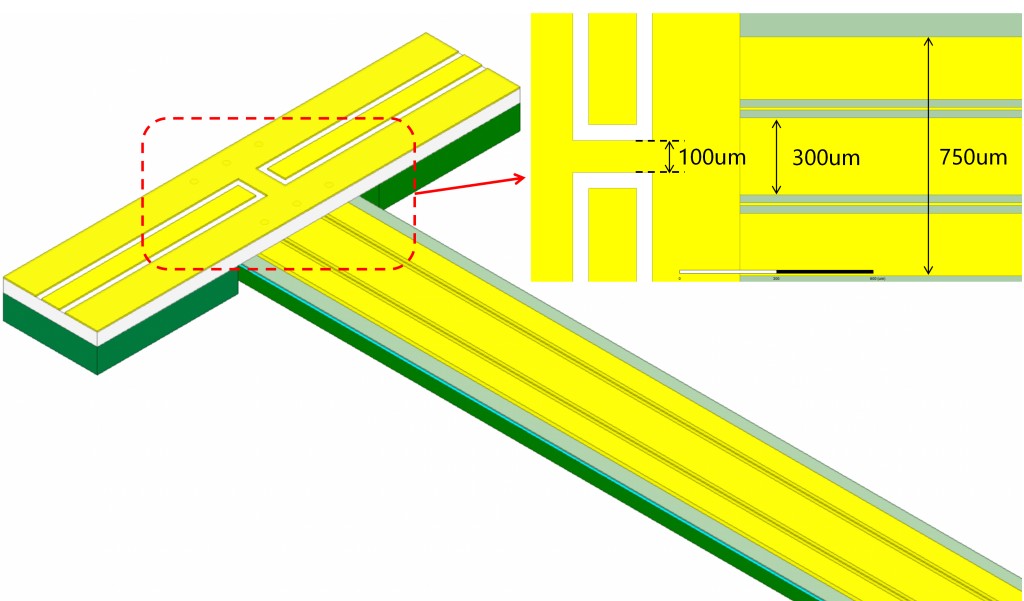

**Figure 7.** Overall structure of dual-channel BBDP structure.

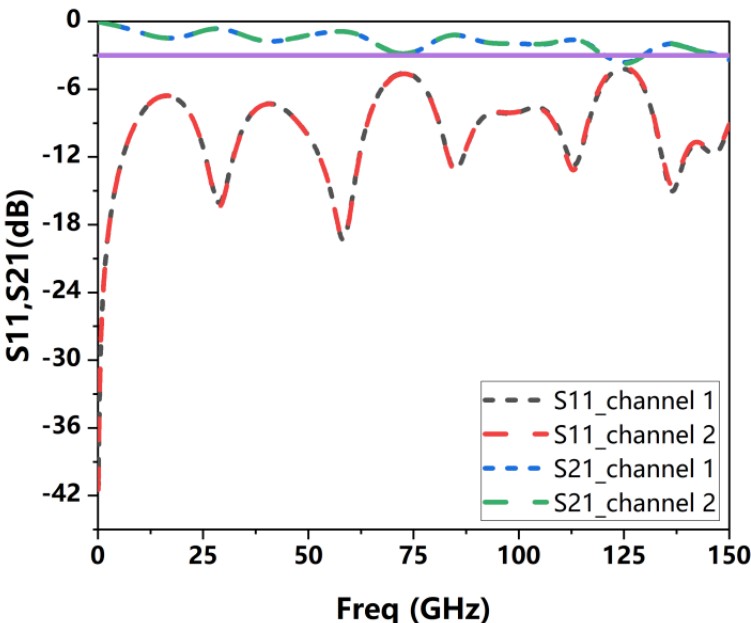

**Figure 8.** S parameter simulation results of the proposed dual-channel BBDP structure.

Due to the symmetrical structure, the S parameter simulation results of the two channels were basically consistent. We selected the S21 parameter of one of the channels for comparison with the S parameter of the single-channel modulator RF interconnection structure, as shown in Figure 9. Basically, the 3 dB S21 measurement in the dual channel was about 120 GHz, which was 25 GHz less than that of the single channel. The main difference was that the amplitude oscillation of the S parameters of the dual-channel structure was more obvious, which might have been caused by the RF coupling between the two pairs of GSG electrodes in the transition chip.

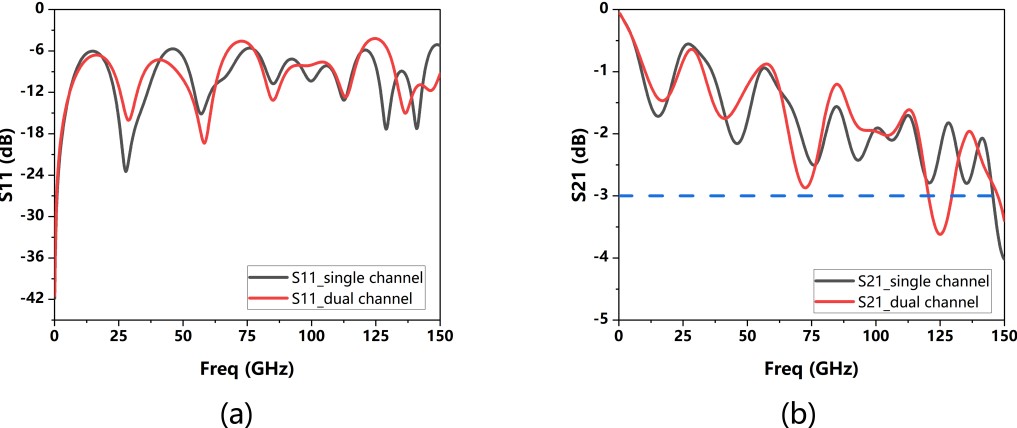

**Figure 9.** (**a**) S11 parameter comparison between single-channel and dual-channel BBDP structure; (**b**) S21 parameter comparison between single-channel and dual-channel BBDP structure.

### 3. Simulation Comparison

In order to compare the above results of the BBDP structure, we also performed the S parameter simulations of the wire-bonded modulator for an RF interconnection structure. We referred to a structure in the literature for simulation [19], as shown in Figure 10. The parameters of the structure are available in the references.

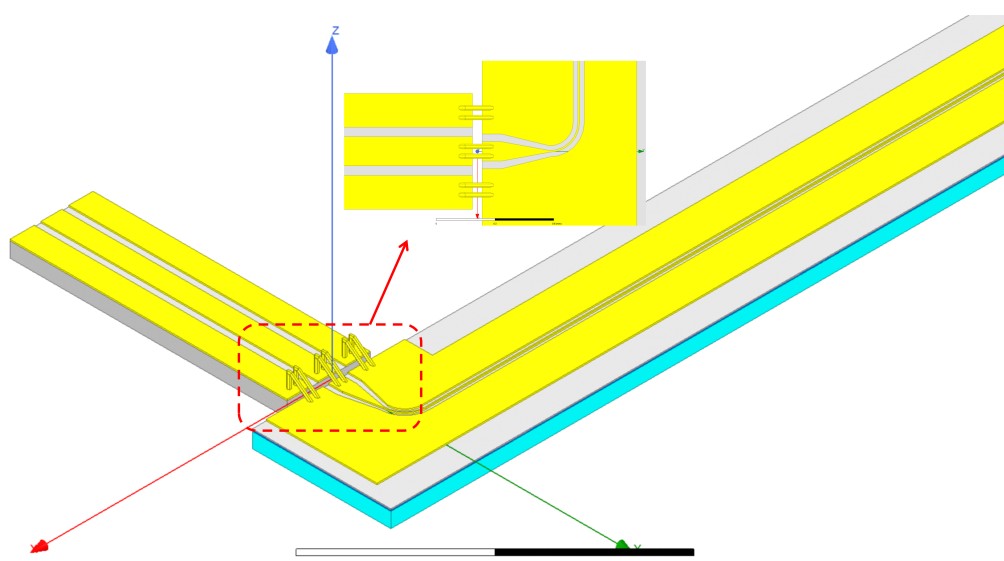

**Figure 10.** Overall structure of the wire-bonded modulator structure (90° curved electrode).

The parameters and simulation settings of the modulator and RF transition chip in the above structure were the same as in the BBDP simulation. The difference was that to wire bond the modulator and the RF transition chip, the electrode structure was bent at a 90° angle. As shown in Figure 10, the bending radius was 180 μm and the tapered transition structure was 250 μm long. The signal electrode in the bonding area was 100 μm wide and the GS spacing (gap between the signal electrode and the ground electrode) was 40 μm.

The simulation results are shown in Figure 11. The 3 dB transmission bandwidth (S21) of this structure was about 95 GHz, which is not better than the bump bonding one. It could be seen that as the signal frequency increased, the transmission performance of the wire-bonded structure began to decline significantly. In addition, the bump bonding structure maintained the same decay rate of transmission performance. From Figures 9 and 11, we can see that the 3 dB transmission bandwidths of the traditional wire-bonding structure and the proposed BBDP structure were about 95 GHz and 145 GHz, respectively. The wire-bonding

technique brougt the unavoidable effect of parasitic capacitance in the interconnection between the RF connector and the microstrip electrodes of the modulator chip. These indicated an improvement in transmission performance for the proposed and designed BBDP structure of approximately 52.6% compared with the wire-bonding structure.

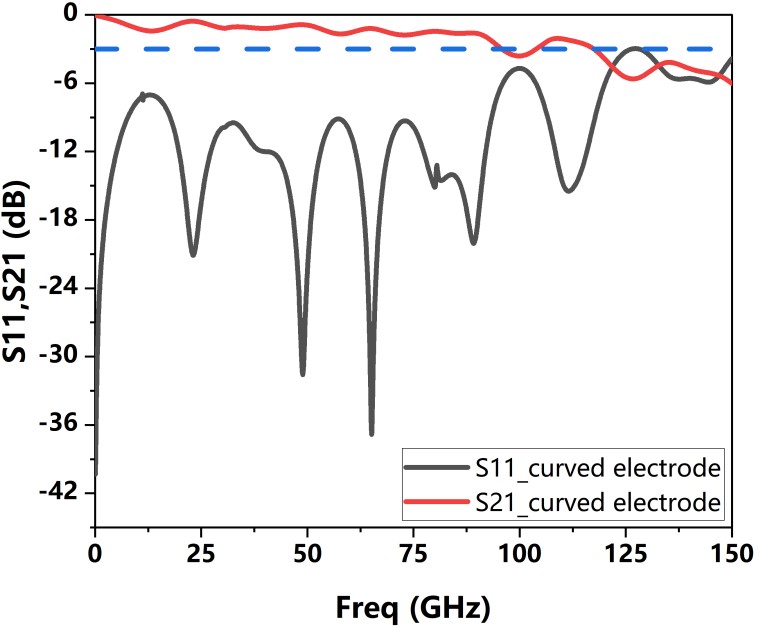

**Figure 11.** S parameter simulation curve of the structure shown in Figure 10.

In addition, considering that the curved design of the modulator electrodes can be avoided by straight-like electrodes [20], we also simulated and compared the RF interconnection structure of the straight GSG electrodes. Figure 12 shows the radio frequency interconnection model of straight GSG electrodes, and its specific structural parameters were consistent with the previous curved electrode structure. Figure 13 shows the S parameter simulation results of this model. The 3 dB transmission bandwidth of S21 was approximately 125 GHz.

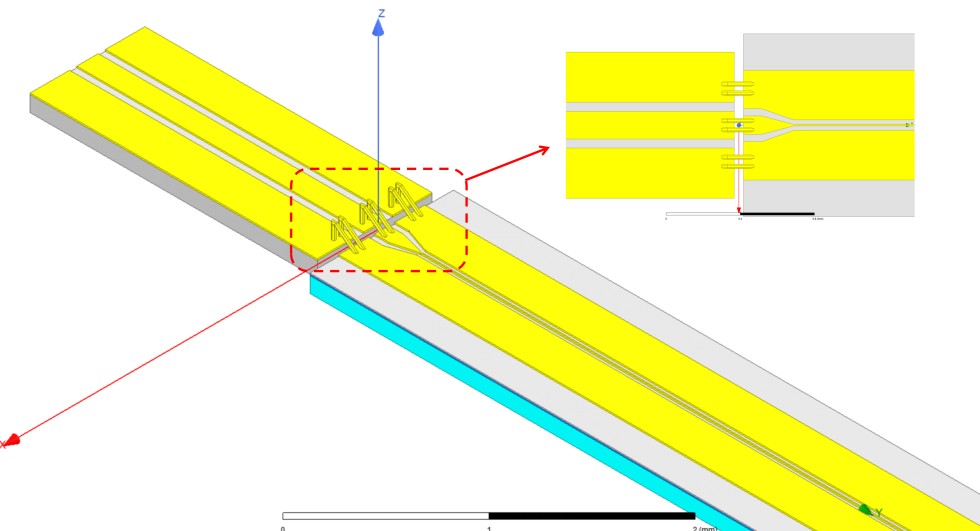

**Figure 12.** Overall structure of the wire-bonded modulator structure (straight electrode).

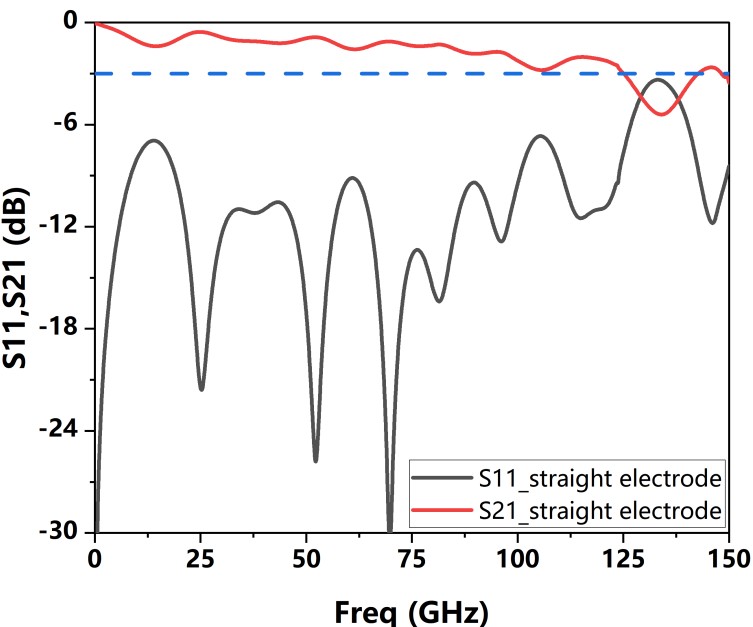

**Figure 13.** S parameter simulation curve of the structure shown in Figure 12.

## 4. Results and Discussion

In the previous simulation calculations, the transmission bandwidth of the dislocation packaging model we designed was approximately 145 GHz. For comparison, the transmission bandwidth of the wire-bonding model with 90° curved electrodes was about 95 GHz, and the transmission bandwidth of the wire-bonding model with straight electrodes was about 125 GHz. Compared with the other two models, the transmission bandwidth of the model we designed increased by 52.6% and 16%, respectively.

There are currently many modulator chips with very excellent performance. However, their packaging generally uses wire bonding, which increases the difficulty of impedance matching during interconnection and leads to a decrease in transmission performance. At the same time, many modulator chips are designed with curved electrodes for transition in order to facilitate packaging, which undoubtedly further increases the transmission loss of the microstrip-like travelling electrodes. For example, the bonded thin film lithium niobate modulator on a silicon photonics platform, designed by P. Weigel's team, can achieve an electro-optical modulation bandwidth of more than 106 GHz, but its electrical transmission bandwidth is approximately between 40 and 50 GHz [21]. A limiting factor is the curved design of its electrode structure. A straight electrode design is adopted to avoid the bending loss of RF transmission. In addition, H. Zwickel's team demonstrated a performance comparison of an electrically packaged silicon–organic hybrid (SOH) I/Q-modulator before and after applying wire-bonding packaging [22]. The 3 dB bandwidth of the modulator before packaging was greater than 65 GHz, but after packaging it was only 21 GHz. These illustrate that wire bonding can induce unavoidable loss for the modulator chip packaging. For the above two issues, the BBDP structure we designed could provide an effective solution, which could improve impedance matching and increase the transmission bandwidth.

Compared to the wire-bonding model of curved electrodes, the BBDP structure has two main advantages. One is that it avoids the design of curved electrodes, and the other is that the transmission loss of bump bonding is smaller than that of wire bonding. Based on the advantages of bump bonding, our model still had a higher transmission bandwidth than the straight electrode model of wire bonding; it showed a 16% improvement compared to the lead package structure using straight electrodes, and the optical waveguide of our structure does not need to be bent and can effectively avoid optical bending losses.

## 5. Summary

We proposed a design for an RF interconnection using bump bonding with a dislocation packaging structure. We used the longitudinal space of the z-axis to realize the misalignment of the modulator and the RF adapter chip, and they are perpendicular to each other on the x–y plane. This eliminated the need for the 90° bent structure of the electrodes, which reduced the parasitic capacitance induced by the wire and its interconnection length. Therefore, the delay in RC resistance and capacitance was reduced, which could effectively improve transmission performance.

To sum up, the dislocated packaging structure we designed has great advantages in improving transmission performance. Compared with the curved electrode model and straight electrode model of wire bonding, the transmission performance of our model was improved by 52.6% and 16%, respectively. The BBDP structure we designed can effectively avoid the bending design of electrodes and can also reduce the bending design of optical waveguides. Meanwhile, the BBDP structure can also take full advantage of bump bonding to reduce transmission losses, since the impedance mismatching between the wire and the microstrip electrodes of the modulator can be removed. We believe that the dislocated packaging structure we designed represents an important alternative path in realizing a high-speed optoelectronic chip, such as an EO modulator, photodetector, etc.

**Author Contributions:** Conceptualization, J.P. and H.L.; Formal analysis, J.P. and H.L.; Funding acquisition, X.W.; Investigation, J.P., L.W., Y.L., R.L., S.D. and H.L.; Methodology, J.P., X.W., H.G. and H.L.; Project administration, X.W., H.G. and H.L.; Supervision, X.W., H.G. and H.L.; Validation, J.P., L.W. and R.L.; Visualization, J.P., Y.L. and S.D.; Writing—original draft, J.P.; Writing—review & editing, H.G. and H.L. All authors have read and agreed to the published version of the manuscript.

**Funding:** This research was funded in part by the Open Project Program of SJTU-Pinghu Institute of Intelligent Optoelectronics (NO. 2022SPIOE103); the National Natural Science Foundation of China (61775084, 62075088), the NSAF (U2330113, U2030103, U2230111), the Youth Talent Support Programme of Guangdong Provincial Association for Science and Technology (SKXRC202304), the Natural Science Foundation of Guangdong Province (2023A0505050159, 2021A0505030036, 2022A1515110970), the Fundamental and application foundation project of Guangzhou (202201010654), and the Fundamental Research Funds for the Central Universities (21622107, 21622403, 21623411).

**Institutional Review Board Statement:** Not applicable.

**Informed Consent Statement:** Not applicable.

**Data Availability Statement:** The data that support the findings of this study are available from the corresponding author upon reasonable request.

**Conflicts of Interest:** The authors declare no conflict of interest.

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
