# Peer review of "RF Interconnection Design of Bump Bonding with a Dislocation Package Structure towards Electro-Optic Modulation Applications"

_photonics, doi:10.3390/photonics10121348_

Round 1

Reviewer 1 Report

Comments and Suggestions for Authors

Reviewer 2 Report

Comments and Suggestions for Authors

The authors propose a design of radio frequency (RF) interconnection of bump bonding with dislocation packaging (BBDP) structure. Through simulation calculations and analysis, the proposed BBDP structure shows a 3dB transmission bandwidth of ~145GHz, 52.6% better than the optimized wire bonding structures (95GHz). the design shows its advantages toward EO modulation with larger electric bandwidth, which can draw the considerable interest from the EO devices community. This paper meets the criteria of the special issue of Photonics, before making the final decision, several points should be addressed as followed:

1Why do you use spherical solder joints for bump bonding?

2Why do the two ground electrodes use the same metal electrode in the GSG electrode structure?

3Can this structural design be expanded to 3 channels, 4 channels or even more?

4There are some typos should be checked through the whole manuscript, such as electrodeof, the axis should be indicated in fig. 1, and the description of the fig. 2 is not clearly.

Reviewer 3 Report

Comments and Suggestions for Authors

The study presents the design of RF interconnection of bump bonding with dislocation packaging structure that poses good significance. However, revisions are required.

1. Separate the related work from the introduction. Provide the related work in a separate section and add more recent and authoritative literature to the reviews.  Define the cons and pros of the existing works and show how the current work has addressed the identified gaps.

2. Highlight the key contributions in section 1 and add the paper structure or organization in section 1 after the contributions.

3. The caption of Table 1 is wrong. Please provide the right caption. The figures are blurred. Authors are to provide sharper figures.

4. The design calculations critical to understanding the underlying principles are missing. Please add the relevant design calculations and the associated proofs. In addition, the authors need to state all assumptions made.

5. The results have not been compared with related work to assess the validity of the proposed design. Also, the implication of the results is missing.

6. The future scope is missing in the conclusion section. Add it.

Comments on the Quality of English Language

Moderate English editing is required. 

Round 2

Reviewer 1 Report

Comments and Suggestions for Authors

N.A.

Author Response

Thanks for your comment. We have provided additional explanations of the experimental procedures and conclusions in the revised manuscript. If there is anything that needs improvement, please leave your comments and we will be happy to continue improving it. Looking forward to hearing from you and thank you again for your comment.

Reviewer 3 Report

Comments and Suggestions for Authors

Dear Authors,

Thank you for addressing some of my earlier comments sufficiently. However, the following comments have not been addressed satisfactorily.

4. The design calculations critical to understanding the underlying principles are missing. Please add the relevant design calculations and the associated proofs. In addition, the authors need to state all assumptions made.

5. The results have not been compared with related work to assess the validity of the proposed design. Also, the implication of the results is missing.

Comments on the Quality of English Language

Minor English editing is required.
